# Therapeutic Advances in Gut Microbiome Modulation in Patients with Inflammatory Bowel Disease from Pediatrics to Adulthood

**DOI:** 10.3390/ijms222212506

**Published:** 2021-11-19

**Authors:** Adi Eindor-Abarbanel, Genelle R. Healey, Kevan Jacobson

**Affiliations:** 1Department of Pediatrics, British Columbia’s Children’s Hospital, Vancouver, BC V6H 3N1, Canada; adi.eindor@cw.bc.ca (A.E.-A.); genelle.healey@bcchr.ca (G.R.H.); 2Division of Gastroenterology, Hepatology and Nutrition, British Columbia’s Children’s Hospital, Vancouver, BC V6H 3N1, Canada; 3BC Children’s Hospital Research Institute, University of British Columbia, Vancouver, BC V5Z 4H4, Canada; 4Division of Gastroenterology, Hepatology and Nutrition, Yitzhak Shamir Medical Center, Affiliated to Tel Aviv University, Beer-Yaakov 7033001, Israel; 5Department of Cellular and Physiological Sciences, University of British Columbia, Vancouver, BC V6T 1Z4, Canada

**Keywords:** IBD, microbiome, microbiome modulation, EEN, prebiotics, dietary fibre, probiotics, antibiotics, FMT

## Abstract

There is mounting evidence that the gut microbiota plays an important role in the pathogenesis of inflammatory bowel disease (IBD). For the past decade, high throughput sequencing-based gut microbiome research has identified characteristic shifts in the composition of the intestinal microbiota in patients with IBD, suggesting that IBD results from alterations in the interactions between intestinal microbes and the host’s mucosal immune system. These studies have been the impetus for the development of new therapeutic approaches targeting the gut microbiome, such as nutritional therapies, probiotics, fecal microbiota transplant and beneficial metabolic derivatives. Innovative technologies can further our understanding of the role the microbiome plays as well as help to evaluate how the different approaches in microbiome modulation impact clinical responses in adult and pediatric patients. In this review, we highlight important microbiome studies in patients with IBD and their response to different microbiome modulation therapies, and describe the differences in therapeutic response between pediatric and adult patient cohorts.

## 1. Introduction

Inflammatory bowel disease (IBD) is a chronic, relapsing inflammatory disease of the gastrointestinal (GI) tract. There are two main IBD subtypes: Crohn’s disease (CD), which affects any part of the GI tract and can result in transmural inflammation and the development of penetrating or stricturing disease, and ulcerative colitis (UC), which traditionally affects only the colon [1]. IBD incidence and prevalence is increasing in a concerning way, particularly in children [2,3,4]. Although disease-onset can occur at any age [5], up to 25% of IBD patients are diagnosed when they are younger than 20 years of age [6].

While the etiology of IBD remains largely unknown, it is thought to arise from altered interactions between the environment and the gut microbiome, which result in an exaggerated immune response in genetically susceptible individuals [7]. Furthermore, early-life events impacting microbiota development, such as maternal diet, dietary behaviours (i.e., breastfeeding versus formula feeding and first solid foods) and antibiotic exposure have been linked to the risk of developing IBD [8].

The alarming rise in the incidence and prevalence of IBD in the industrialized world suggests that genetic factors only play a partial role in the development of IBD, and rapid changes in dietary and other environmental factors likely significantly contribute to this rise [9]. Moreover, under germ-free conditions (no microorganisms) animal models of IBD do not develop disease or the disease is significantly attenuated, suggesting that intestinal microbes are essential for the development of intestinal inflammation in IBD [10].

To date, most of the treatments for IBD are based on systemic immunomodulation or targeting specific cytokines in the inflammatory cascade. Current treatments for IBD are not universally effective and have the potential for adverse events. Therefore, there is emerging interest in treatments that positively impact the gut microbiome as an effective strategy to prevent and treat IBD.

For the past two decades, numerous studies have demonstrated the crucial role the gut microbiome plays in the development and maintenance of intestinal health, but also in intestinal inflammation [11]. In a healthy state, the intestinal microbiome works symbiotically in several important roles such as maintenance of the gut epithelium, production of vitamins, nutrient metabolism and interactions with several key immune system signaling pathways [12]. One of the key roles of several bacterial species is the fermentation of non-digestible carbohydrates into short-chain fatty acids (SCFAs) [13]. SCFAs also have a role in cell immunity and as a source of energy for colonocytes (reviewed in more detail in a later section).

The most highly represented bacterial phyla in the intestinal tract are *Firmicutes*, *Bacteroidetes*, *Proteobacteria* and *Actinobacteria* [14]. In healthy individuals, Firmicutes and Bacteroidetes account for about 90% of the total microbiota [15]. Notably, Firmicutes and Bacteroidetes, in conjunction with oligosaccharide-fermenting bacteria such as *Bifidobacterium spp.*, are able to produce SCFAs by fermenting dietary plant fibre [16]. According to published data comprising research undertaken mostly on adult patients, the intestinal microbiota of patients with IBD is characterized by an increased abundance of *Proteobacteria* and a decrease in members from the *Firmicutes* and Bacteroidetes phyla compared to healthy individuals [17]. There is also a reduction in microbiome diversity during a state of inflammation [18]. Notably, these changes can be demonstrated in the same patient in different locations in the intestine [19]. Furthermore, some studies have demonstrated a decreased abundance of SCFA producing species, such as *Faecalibacterium prausnitzii* in CD [20] and *Roseburia hominis* in UC [21] patients, which likely further promotes a pro-inflammatory state. A decrease in specific SCFAs has also been noted in fecal samples of IBD patients [22].

Gut dysbiosis occurs when there is an imbalance in the microbiota ecosystem disrupting normal microbial functions. In this review, we focus on the different therapeutic modalities introduced to correct the dysbiosis observed in both pediatric and adult patients with IBD. Since there are known differences between gut microbiota and interventions according to age, we reviewed the literature comparing microbiome modulation among pediatric and adult patients.

## 2. Differences and Similarities in Microbial Modulation of Pediatric Versus Adult Patients with IBD

The modulation of gut microbiota was already common practice in IBD, even before the era of microbiome research, via antibiotic and nutritional therapies. Many of these treatments were used after their empirical success, without knowing the exact mechanism or treatment target. Characterizing specific gut microbiota changes provided an additional route of investigation into the etiology and treatment of pediatric IBD.

The dysbioses IBD patients experience are thought to influence a variety of functions that are meticulously orchestrated by the gut microbiota, such as fermentation of dietary fibres [23], pathogen defence [24], synthesis of vitamins [23] and drug metabolism [25], as well as a fundamental role in promoting immune maturation [26] and metabolic homeostasis [27].

Growing evidence suggests that changes in the metabolites produced by the gut microbiota as well as microbial metabolic pathways explain some of the causal mechanistic relationships linking the gut microbiota and IBD. Moreover, these insights may result in new innovations regarding microbiome-targeted therapeutic interventions [28]. The Kyoto Encyclopedia of Genes and Genomes (KEGG) can be used to identify and reconstruct genes into broad biological pathways via shotgun metagenomic sequencing data. Metabolite groups that are of interest include SCFAs, bile acids and tryptophan metabolites, as they play an essential role in normal immune development, intestinal homeostasis and IBD [28].

Therefore, modulating the composition of gut microbiota in IBD patients or the metabolic pathways and metabolites produced could have a beneficial impact on inflammatory pathways and on the patient’s gut health.

Although the gut microbiome after three years of age is considered more “adult- like” [29], with a stable “signature” already established [30,31,32], there is ongoing adaptation to this ecosystem. Ringel-Kulka et al. investigated the differences in the microbiomes between healthy children and adults using high throughput microarray analysis. They demonstrated that children have a 3.5-fold greater abundance of *Bifidobacterium* spp. than adults and had a less diverse microbiota [33].

Conversely, it seems that the microbiota changes in pediatric IBD are similar to what was previously reported in adults. In previous studies in adult patients, CD patients had a decreased abundance of Actinobacteria and Bacteroidetes, and an increased abundance of Proteobacteria [17,34]. Similar results were demonstrated in pediatric patients [35,36,37,38,39]. In addition, when comparing between CD patients stratified by age, no systematic changes were found with different ages of diagnosis, suggesting that CD-associated dysbiosis is already established in younger CD patients [40].

On the other hand, the IBD phenotype in pediatric patients is considerably different from adults. Children with IBD have more extensive disease and exhibit a more severe disease course, and in very early onset (VEO) disease the whole colon is typically involved [41,42,43]. Additionally, microbiota modulation has been a mainstay of the pediatric IBD treatment repertoire and has been much more extensively utilized than in adult practice. Nutritional therapy with exclusive enteral nutrition (EEN) is one of the leading induction therapies in pediatric CD [44] and has only been used in select adult patients, with some evidence of a weaker efficacy [45,46].

The different microbiota profiles between children and adults in the general population, and the altered response to nutritional therapy between adult and pediatric patients, raise the concern that results from adult microbiota research cannot be easily extrapolated to pediatric patients. Therefore, there is a need to investigate these nutritional interventions, as well as other microbiome modulating therapies, specifically in a pediatric population.

In the next section we aim to provide an overview of the different gut microbiome modulation therapies that have been investigated in pediatric patients with IBD compared to adult IBD patients.

### 2.1. Nutritional Therapies

#### 2.1.1. Exclusive Enteral Nutrition (EEN)

EEN has been advocated as a first line therapy for pediatric CD since the 1990s, more than a decade before the evolution of high throughput sequencing and extensive gut microbiome research. EEN provides 100% of calorie and nutrient requirements via liquid formulations delivered orally or through nasogastric or gastrostomy tubes for between 6 and 8 weeks [47]. This first-line, steroid-sparing therapy in pediatric CD results in remission rates of 60–80% [48,49,50]. Interestingly, EEN efficacy is independent of the formula types (i.e., polymeric versus elemental formula) or the administration route [51,52]. Although EEN is considered highly effective, the mechanism of action is not completely understood. Recent metagenomic and metabolomic research is focused on identifying pathways that lead to the response. Since EEN is widely used in pediatric patients, it is no wonder that most of the data available on microbiome modulation during EEN treatment were generated in pediatric patients, with studies in adults with IBD lacking.

The first study to investigate changes in gut flora as a potential mechanism of EEN was published in 2005 [53]. Lionetti et al. followed a small group of nine children with CD that received a course of EEN (Modulen IBD, Nestle, Vevey Switzerland) for eight weeks, and investigated the changes in their microbiome using 16S ribosomal DNA polymerase chain reaction and temperature gradient gel electrophoresis, compared to healthy controls. EEN-induced remission was associated with profound modifications of the band profile corresponding to different bacterial species of the fecal microflora, but the specific changes could not be elucidated using these methods.

A later study [54] demonstrated paradoxical results, characterized by a reduced diversity and a decreased abundance of *Bacteroides* genus and *Clostridium coccoides* species in children treated with EEN. Several small studies, summarized in a recent review, demonstrated conflicting results [55]. Most studies show that EEN is correlated with a reduction in bacterial diversity, creating a community structure even more dissimilar than that of controls [56]. These results are puzzling, adding more to the mystery of the mechanism/s of action of EEN. One of the most recently conducted randomized clinical trials [57] compared clinical remission, mucosal healing and bacterial composition between pediatric patients treated with EEN and corticosteroids. They demonstrated that although patients treated with EEN had a significantly higher proportion of mucosal healing, steroid-treated patients had higher abundance of butyrate-producing bacteria, which provides further paradoxical results.

Studies on the impact of EEN on microbial metabolites also demonstrate results that are likely contradictory to what might be expected from a beneficial therapy. Gerasimidis et al. [58] collected stool samples from pediatric CD patients during and after EEN treatment. Surprisingly, butyrate, which is considered a beneficial SCFA, decreased during EEN. These paradoxical results might reflect an opposite causality. It is possible that these observed changes are because EEN contains very few dietary components, when compared to a regular diet, and no complex carbohydrates. Therefore, the reduced alpha diversity, lower abundance of fibre-degrading bacterial taxa and low levels of SCFA might be simply due to a lack of fermentable substrates. However, Quince et al. [56] demonstrated that, although pediatric CD patients demonstrated higher levels of genetic functional diversity prior to EEN treatment compared to controls, the levels tended to decrease during EEN to levels similar to controls. They postulated that this depicts a greater range of functional roles that can be exploited by the microbiota in an inflamed gut and, therefore, these changes might be the reason for the beneficial effect of EEN. Furthermore, a recent comprehensive investigation [59] comparing the metabolic products of pediatric CD patients that responded to EEN treatment to those who did not found that the fecal metabolome of responders and non-responders differed prior to commencing EEN therapy. A specific fecal metabolic profile was able to predict EEN responses prior to treatment, with an AUC of 0.8. Specifically, cadaverine and trimethylamine were found in greater concentrations in CD patients, decreased during EEN and normalized in responders only.

Data on the impact of EEN on the gut microbiota in adult CD patients are scarce. One older [60] study in a small patient cohort (*n* = 8) compared the microbiome of patients treated with elemental formula to those treated with total parenteral nutrition, and found that elemental formula did not reduce species diversity. Since EEN is not used routinely in adult patients as induction therapy, Costa-Santos et al. [61] investigated microbiota changes in adult CD patients treated with EEN as a means of improving nutritional status prior to intestinal resection due to perforating or stricturing disease. The patients were treated with EEN for only two weeks, followed by six months of partial enteral nutrition. They observed microbial composition changes after EEN, which were characterized by a significant decrease in alpha diversity as well as in the *Enterobacteriaceae* family. However, EEN did not affect postoperative recurrence or gut microbiota composition six months following surgery. Due to the unique patient population studied, it is unclear if these results are attributed to the EEN therapy or the intestinal resection.

#### 2.1.2. Prebiotics and Dietary Fibre

The International Scientific Association for Probiotics and Prebiotics (ISAPP) defines prebiotics as ‘a substrate that is selectively utilized by host microorganisms conferring a health benefit’ [62].

The most widely used fibres in gastrointestinal diseases are non-digestible soluble fibres that are fermented in the colon and thus increase the concentration of beneficial bacterial metabolites such as SCFAs [63]. Soluble fibre, specifically fructooligosaccharides, inulin and galactooligosaccharides are found naturally in foods and are also added as a food ingredient in various commercial products to promote gut health. They have been shown to benefit gut health, not only indirectly via modulation of the gut microbiome, but also in a direct anti-inflammatory manner via TLR4 binding on intestinal epithelial cells [64]. A recent review, published by our group [65], outlined the different properties and significance of soluble dietary fibre in the context of IBD.

Several observational studies have investigated the implications of fibre consumption during childhood and adolescence and subsequent development of IBD, and found that fibre intake, particularly from vegetables and fruits, protects against CD, but not UC, development [66,67]. These food products contain mostly soluble fibres that are fermented in the colon and, therefore, increase the concentration of beneficial microbial metabolites such as SCFA [65]. Dietary patterns with high consumption of meats, fatty foods and desserts (i.e., Westernized dietary pattern) were associated positively with CD development. Therefore, it is difficult to extrapolate whether the high consumption of fibre has beneficial effects in isolation or in combination with a decreased overall Westernized dietary pattern.

Although the investigations on the association between fibre consumption and reduced prevalence of UC were disappointing, there is growing evidence that treating adult UC patients with inulin-type fructan prebiotics can have clinical benefits. Furrie et al. [68] demonstrated that synbiotic therapy with *Bifidobacterium longum* combined with inulin-oligofructose significantly reduced inflammatory cytokines in adult UC patients. In another small randomized-controlled trial, Valcheva et al. [69] compared the clinical effects of two doses of inulin-type fructans in adult UC patients with mild to moderate disease on no medications or 5-Aminosalicylic acid (ASA). They demonstrated that after nine weeks of treatment there was a dose dependent decrease in the abundance of *Bacteroidetes* in biopsy samples. Additionally, the prebiotic increased colonic butyrate production in the 15 g/d dose group, with fecal butyrate levels being negatively correlated with Mayo score (*r* = −0.50; *p =* 0.036).

Currently, there are no published randomized controlled trials investigating prebiotics’ modulating potential on the gut microbiota in pediatric patients with IBD. However, a placebo-control study [70] in healthy children aged three to six years, supplemented with inulin-type fructans or maltodextrin, demonstrated that prebiotic supplementation resulted in selective modulation of the gut microbiota, with a higher abundance of *Bifidobacterium* spp. in children receiving the prebiotic.

Our group demonstrated [71], using a T cell transfer mouse model of colitis, that EEN enriched with inulin-type fructans (EEN IN) could be a novel therapeutic option. Mice treated with EEN IN had a reduction in colitis, higher fecal butyrate concentrations and *Bifidobacterium* spp., and lower potentially pathogenic bacterial species compared to mice treated with EEN alone. These findings will be followed by a randomized-controlled study in pediatric IBD patients that will hopefully help reveal the true impact prebiotics have on clinical outcomes and microbiota modulation in pediatric patients with IBD.

### 2.2. Probiotics

Probiotics are defined by ISAPP as “live microorganisms that, when administered in adequate amounts, confer a health benefit to the host” [62,72]. Many probiotics are derived from the commensal gut microbiota of healthy individuals and aim to displace the dysbiotic microbiota that fails to protect the gut during inflammation. Their role is to mimic the bacteria found in a “healthy” gastrointestinal ecosystem and to create a homeostatic effect. The different roles of specific bacterial species in immune regulation have been previously outlined in a review by Hevia et al. [73].

Probiotics are widely used by both adult and pediatric IBD patients due to their high safety profile. It is perceived by patients as being a more “natural” treatment. Probiotics are also considered as part of the mainstay treatment for the prevention of pouchitis, a complication that is common in UC patients post colectomy, and formation of ileal pouch-anal anastomosis [74]. A recent review [75] summarized the influence of probiotics on the gut microbiome. It appears that bacteria during probiotic treatment can survive transit through the digestive tract, however, in several clinical trials it did not seem to change the diversity or composition of the gut bacteria community [76,77]. Additionally, with the cessation of treatment any beneficial effect on the host microbiota seems to be lost [78]. However, there is some evidence to suggest that yogurt containing *Bifidobacterium animalis* can increase the level of SCFA producing bacteria which can in turn influence systemic metabolism and energy expenditure [79]. Interestingly, it seems that low abundant species are more likely to expand in the host luminal tract than those already present in high abundance [80].

It is important to note that there is a plethora of different probiotic preparations available. These vary in the specific strains used, the number of strains in a single preparation, the dose of probiotic in the regimen and the form of the preparation. The strains of interest that are usually investigated are *Escherichia coli (E. coli)* strain Nissle 1917 [81], *Lactobacillus reuteri* [82], *E. Coli* (serotype 06:K5:H1) [83], *Bifidobacterium* 536 [84] and *L. casei* strain ATCC PTA-3945 [85] and the yeast *Saccharomyces boulardii* [86]. The most frequently investigated combination probiotic used in IBD is VSL#3, which includes four strains of lactobacilli (*Lactobacillus casei, L. plantarum, L. acidophilus* and *L. delbrueckii subsp. bulgaricus*), three strains of bifidobacteria (*Bifidobacterium longum, B. breve* and *B. infantis*) and *Streptococcus salivarius subsp. thermophilus*.

A recent Cochrane review assessed the effectiveness of probiotics compared with placebo or standard medical therapy to induce remission in patients with active UC [87]. The review included twelve studies with adult participants and two studies with pediatric participants with mild to moderate UC. It was concluded that probiotics are able to induce clinical remission compared to placebo (RR 1.73, 95% CI 1.19 to 2.54), and one study demonstrated slightly better efficacy with combined probiotic and 5-ASA therapy compared to treatment with 5-ASA alone. These studies used a variety of probiotic regimens with different combination therapy and a variety of modes of administration.

Since most physicians will be reluctant to use probiotics instead of conventional therapy to achieve remission, it may be more helpful to assess whether probiotics have a role in maintaining remission combined with conventional therapy. According to the most recent European Crohn’s and Colitis Organization (ECCO) and European Society of Pediatric Gastroenterology, Hepatology and Nutrition (ESPHGAN) guidelines from 2018, probiotics may be recommended as a complementary therapy for adults and children with mild UC, but not as a first-line therapy [88].

The data on the utilization of probiotics in CD are limited, but the scarce data that are available suggest that probiotics are ineffective. Bejarnason et al. [89] investigated the influence of probiotics on both adult UC and CD patients in remission or with mild symptoms. They used a combination of probiotics that included *Lactobacillus rhamnosus*, *Lactobacillus plantarum, Lactobacillus acidophilus* and *Enterococcus faecium* and found that in the UC patients, but not CD patients, fecal calprotectin (Fcal) levels were significantly lower after taking the regimen compared to the controls. There was, however, no change in clinical symptoms in either group.

Although the use of probiotics is also recommended in pediatric patients with UC, the data on this age group are very limited. *Lactobacillus* GG was investigated in pediatric CD patients as a complementary therapy to standard medications, and was found not to be effective in preventing relapse [90]. In children with UC, on the other hand, the probiotic VSL#3 was considered to be successful as a concomitant therapy both for disease induction and maintenance [91]. Data on the effect probiotics have on the gut microbiota of pediatric patients with IBD are lacking. Furthermore, these regimens were only investigated in small randomized controlled or clinical trials and, therefore, need to be assessed further [91].

### 2.3. Antibiotics

Antibiotics and IBD have a complex relationship. On the one hand, antimicrobial substances can have a hazardous effect on the homeostasis of the host microbiota, leading to a community shift characterized by increased *Enterobacteriaceae* and reduced *Clostridia* abundance, which is considered a possible pre-IBD state [92]. Additionally, IBD patients treated with antibiotics are at high risk of developing an overgrowth of pathogenic bacteria (*Clostridioides difficile*), fungi (candida) and bacteriophages [93].

Indeed, antibiotics have been an integral part of the treatment repertoire in both pediatric and adult IBD, even prior to the era of immunomodulation and biologic treatment. They have been used widely, mostly in special conditions such as pouchitis, perianal disease and abdominal abscesses, but also in uncomplicated luminal disease. Several potential mechanisms have been suggested for the role of antibiotics in treating IBD [94]. Firstly, antibiotics can have a direct influence on the luminal gut microbiota, favoring flora that are associated with anti-inflammatory properties, such as *Bacteroides* and Firmicutes, and reducing microbes that are associated with inflammation, such as *Enterobacteriaceae*, including *Escherichia coli* and *Fusobacterium* [17]. Antibiotics can also modify metabolic enzymatic pathways generated by gut bacteria [95,96]. In addition, in CD, where there is evidence of pathobionts invading the mucosa [97], antibiotics can have a role in targeting these specific species. However, antibiotics are mostly used empirically, without locating a specific microbial treatment target.

Although there are no data comparing pediatric and adult patients with IBD directly, there is scarce evidence that young pediatric patients might experience an even greater effect from using antibiotic treatment as a maintenance therapy. A case series [98] with Very early onset IBD VEOIBD patients with a mean age of 1.6 years demonstrated that oral treatment with Vancomycin with or without Gentamycin can induce sustained remission in VEOIBD patients that were refractory to other treatments. However, data on the characteristics of the microbiome of VEOIBD patients are lacking, and therefore, the mechanism of action for this approach is still unknown.

Only recently, with the development of high throughput microbiome techniques, do we have a better understanding of the role that stool and tissue associated microorganisms play in IBD patients treated with antibiotics. Furthermore, there is evidence that the specific gut microbiome in patients with IBD will respond better to antibiotics; therefore, by assessing patient stool samples prior to treatment, we can choose the right candidate/s for anti-microbial therapy [99]. The main goals of antibiotic treatment should be to target specific pathobionts or to achieve favorable microbiome or metabolome modulation. However, although data using this approach are emerging, they are still very limited (Table 1).

### 2.4. Fecal Microbiota Transplantation (FMT)

FMT was first reported in 1958 for treating refractory and recurrent *Clostridiodes diffcile* infection (RCDI) [103], and was validated in the past decade as an effective therapy, with over a 90% success rate [104]. Since FMT has been shown to be an effective and safe treatment, it was listed in both American and European guidelines as an official treatment for RCDI [105,106]. The beneficial effect of FMT for RCDI led researchers to explore this treatment option in IBD. Since Bennet [107] reported the first case of FMT in a patient with UC in 1989, additional data, including randomized controlled trials, have emerged to investigate the role of FMT in IBD. Most studies have been undertaken in adult IBD patients, but some studies included patients in pediatric age groups [108]. However, in the past five years, only randomized controlled trials testing the efficacy of FMT in adult IBD patients have been published (Table 2). Two study protocols were recently published describing studies investigating the effects of FMT in pediatric CD and UC patients [109,110], but results are not available yet.

Although the published data on FMT in IBD patients look promising, with some studies reporting clinical remission rates of 20–30% compared to standard treatment alone [111,112], many of these results were not statistically significant due to small sample sizes. Furthermore, there is large heterogeneity in the cohorts of participants studied and the FMT protocols used, therefore, it is difficult to conclude what the most appropriate patient population to treat is or the best FMT methodology to follow.

Some studies used fresh fecal samples for the transplantation, while others used frozen. For some studies, one healthy donor was used for all participants, whereas in other studies, a pooled fecal sample from several donors was used. More recently, there has been growing interest in autologous fecal transplantation as the preferred method, due to better safety profiles [113]. There is also no unified protocol on the preferred route of fecal administration. Most of the studies use colonoscopy to administer the FMT, but others debate that gastroscopy is an equally effective method [114], while other groups use oral capsules [115].

The patient cohorts recruited for these studies are also very diverse. In most studies, the participants were treated with different induction or maintenance therapies, which can also affect the results. Furthermore, in studies conducted in patients with UC, those with isolated proctitis are generally excluded; however, in studies performed in CD patients, there are often no unifying disease location inclusion or exclusion criteria.

Another limitation of FMT studies is that long term follow-up of study participants is often lacking. In most studies, participants that were randomized as controls were permitted to proceed with fecal transplantation after the end of the study observation period, which limits the ability to compare long term outcomes between groups.

For these reasons, it is difficult to extrapolate whether FMT is an appropriate alternative microbiota modulation treatment in adult IBD patients. As very few studies have been conducted in the pediatric IBD population, additional data need to be generated before any valid conclusions can be drawn in this population. It is worth noting that researchers wanting to undertake FMT interventions in pediatric cohorts should consider the added adverse risks associated with performing endoscopies under anesthesia in this vulnerable group. Additionally, alternative FMT delivery methods, such as fecal capsules, may not be feasible, as the large capsules may be impossible for younger children to swallow.

**Table 2 ijms-22-12506-t002:** Randomized controlled trials investigating microbiome changes after FMT in adult and pediatric IBD patients.

Study	Type	*n*	Age	Severity	Route	Donor	Type of FMT	Clinical Response	Change in Microbiome	Follow Up
Sokol 2020 [111]	CD- Colonic and ileocolonic	8	18–70	Originally HBI > 4, but post remission induction with steroids	Colonoscopy Single dose	Healthy donors age 20–50	Fresh	25% difference between FMT to control (NS)	-No significant changes in donor microbiota between those who responded and those who did not -Outcome of >60% colonization of the donor microbiota at 6 weeks was not achieved	24 weeks
Paramsothy 2017 [112] (FOCUS study) Paramsothy 2019 [116] (microbial analysis of FOCUS study)	UC	41	18–75	Mayo score 4–10 IBDQ score 123 (99–157)	Initial colonoscopic and then intensive multidonor FMT enemas 5 d/wk for 8 weeks	Blended homogenized stool from 3–7 unrelated donors	Fresh	Primary endpoint of steroid-free clinical remission together with endoscopic remission or response at week 8. 27% (11 of 41) of patients assigned fecal microbiota transplantation met the primary endpoint, compared with 8% (three of 40) of those allocated placebo (*p* = 0.021)	Patients in remission after FMT had enrichment of *Eubacterium hallii* and *Roseburia inulivorans* and increased levels of short-chain fatty acid biosynthesis and secondary bile acids. Patients who did not achieve remission had enrichment of *Fusobacterium gonidiaformans, Sutterella wadsworthensis*, and *Escherichia* species and increased levels of heme and lipopolysaccharide biosynthesis	24 weeks
Costello 2019 [117]	UC	38	≥18	total Mayo score of 3 to 10 points and an endoscopic subscore of ≥2	Colonoscopy followed by 2 enemas over 7 days	Anaerobically prepared pooled donor FMT or autologous FMT	Frozen	OR of steroid free remission of 5 (1.2–20.1) *p* = 0.03. Endoscopic remission was NS	Increased abundance of *Anaerofilum pentosovorans* and *Bacteroides coprophilus* species was strongly associated with disease improvement following donor FMT Changes in SCFA were not significant	8 weeks and 12 months (but by 12 months most of the patients had donor FMT)
Yang 2020 [114]	CD	27	18–60	Mild to moderate CDAI > 150	Randomized to colonoscopy vs. Gastroscopy A second FMT 1 week afterwards	Healthy donors	Fresh	No significant differences were seen between the gastroscopy and colonoscopy groups (clinical response, 76.9% and 78.6%, respectively; clinical remission, 69.2% and 64.3%, respectively	Only investigated changes between patients and donors not between study groups	8 weeks
Schierova 2020 [118]	Left sided UC	8	Median 40 IQR (31–66)	Median mayo score of 5.5	5 enemas administered at the first week, then once a week until the end of 6th week	1 healthy donor	Frozen	Clinical response was 62.5% in both FMT and 5-ASA groups	*Faecalibacterium*, *Blauti, Coriobacteria*, *Collinsela*, *Slackia*, and *Bifidobacterium* were significantly more frequent in patients who reached clinical remission	12 weeks
Sood 2019 [119]	UC	31	Mean age 33 SD ± 12.4	Clinical remission	Colonoscopy every 8 weeks for 48 weeks	1 healthy donor	Frozen	NS in clinical remission Endoscopic remission in 58.1% of FMT and 26.7% in placebo *p* = 0.026	No microbiome analysis	48 weeks
Crothers 2021 [115]	UC	6	≥18	Mayo score between 4 and 10	Colonoscopy and then 12 weeks of daily Encapsulated oral FMT	2 healthy donors	Frozen	NS	FMT lead to community-level changes in the gut microbiota creating measurable similarity (beta diversity, Jensen-Shannon divergence index) between FMT subjects and their donor. *p* < 0.01	36 weeks

UC—ulcerative colitis, CD—Crohn’s disease, HBI—Harvey Bradshaw index, CDAI—Crohn’s disease activity index, FMT—fecal microbial transplantation, NS—not significant, SD—standard deviation, FMT—fecal microbiota transplantation, OR—odds ratio, ASA—amino salicylic acid.

### 2.5. Postbiotics

As stated above, SCFA are microbial metabolites that exert beneficial effects on the intestinal mucosa. Usually, they are produced by beneficial gut microbiota due to fermentation of prebiotics and other non-digestible dietary fibres. The main SCFAs known to promote gut health are acetate, propionate and butyrate [120]. The different roles SCFAs play in cell immunity and their anti-inflammatory properties have been gradually revealed over recent years. SCFAs, particularly butyrate, are used as a source of energy by intestinal epithelial cells. In addition, these molecules act as a link between the gut microbiota and underlying immune system, and modulate different aspects of the immune system such as immune cellular function and innate mechanisms of host defense [121]. They also have a role in activating specific anti-inflammatory cascades [122], therefore, their role has been extensively investigated in IBD patients.

In previous sections, we reviewed specific studies which reviewed modulation of the production of SCFAs by altering the composition of the gut bacteria or the fermentable substrates provided. Another approach that is being investigated to increase concentrations of luminal SCFAs is via topical or oral administration of the SCFAs themselves.

Two decades ago, Steinhart et al. [123] attempted to treat left-sided adult UC patients with butyrate enemas, but found the therapy to be non-efficacious. A more recent randomized control trial [124] investigated supplementation with ester propionyl-l-carnitine (PLC), a source of L-Carnitine that is required for the transport of activated fatty acids into the mitochondria for beta-oxidation. They found that adult UC patients on a stable dose of 5-ASA or thiopurines that were concurrently treated with PLC had higher remission rates, especially in the high dose group. A study that recruited adult CD patients with mild to moderate disease showed that oral butyrate treatment was also efficacious, but they did not include a placebo group in their study [125].

Recently, researchers used a novel approach, which took into consideration the low diffusion capacity of oral and enema delivered SCFAs. They created lipophilic butyrate-containing microcapsules to provide enhanced capacity for intestinal diffusion and facilitate slow release of the active ingredient [126]. For the first time, it was demonstrated that butyrate delivered this way could alter the gut microbiota of IBD patients by increasing the bacteria able to produce SCFAs in both UC and CD patients. There was no significant change in clinical symptoms, but some significant effect on subjective quality of life in UC patients was observed. Although these results are promising, there are still insufficient data to support this type of treatment in adults, and no studies have been published in pediatric patients with IBD.

Other metabolic derivatives were also investigated as contributing factors in the pathogenesis of IBD. Adult CD patients were found to have altered metabolism of Tryptophan [127], and fecal bile acids were repeatedly investigated in IBD patients [128,129]. Notably, in a small cohort of IBD patients and healthy controls, Duboc et al. reported that the microbial dysbiosis in IBD patients was associated with alterations in the luminal bile acid pool, potentially eliminating the beneficial anti-inflammatory effects of some bile acids [130]. However, studies on these metabolites and others are in the early stages of evaluation. With the advancement of multi-omic investigations, there is no doubt that their role in IBD pathogenesis and management will be revealed in the near future.

## 3. Conclusions

Modulation of the gut microbiota to reverse microbial dysbiosis represents a promising direction in the treatment of IBD (Table 3). Even with advancements in high-throughput microbiome analysis, the exact of mechanisms of action is still unclear, with respect to the impact microbiome modulation is having on IBD outcomes. Some of the proposed mechanisms include suppressing an inflammatory state, reducing invading pathobionts and promoting gut epithelial cell repair. These effects can be achieved using a variety of methods, as discussed above.

The usual pipeline for medication approval for patients with pediatric IBD is demonstrating extensive clinical benefits in adults first before implementing a new treatment in pediatric patients. Although the rationale for this approach is understandable, to ensure a reasonable safety profile, it exhibits a challenge in IBD. Pediatric IBD patients are not “small adult IBD patients”. Pediatric patients display a distinctive disease profile compared to adults, usually presenting with more extensive disease, most commonly pancolitis, with isolated ileal disease being an unusual presentation [126,131]. They also seem to exhibit different responses to treatment modalities, with EEN being a leading induction treatment in pediatric IBD only [46], as well as better response to antibiotic treatment [98], particularly in VEOIBD patients.

Although there is growing evidence to suggest differing responses to microbiome modulation in pediatric versus adult patients, the reason for this is still unclear. There are only scarce data on the different microbiome changes initiated after nutritional, probiotic and antibiotic treatment, despite these treatments having been implemented in pediatric IBD medicine for several decades. The evidence on FMT and postbiotics is still lacking.

EEN is the most extensively investigated treatment in pediatric IBD patients with regards to disease outcomes and modulation of the microbiome, however, the data are drawn mostly from small studies, at times providing conflicting results. Moreover, the published data have not shed light on the exact mechanism of action for this treatment modality.

The data in adult patients with IBD are also not robust enough to suggest a clear medical approach using the previously discussed microbiome-targeted methods. Although modulation of the microbiome to enhance disease outcomes for both adult and pediatric patients with IBD shows promise, there is still a pressing need to undertake large, well-designed trials to better elucidate the mechanisms of action and the treatment options that are likely to have the greatest impact for IBD patients of all ages.

## Figures and Tables

**Table 1 ijms-22-12506-t001:** Randomized controlled trials investigating microbiome changes due to antibiotic treatment in adult and pediatric IBD patients.

Study	Type	*n*	Age	Severity	Antibiotics	Clinical Response	Type of Analysis	Change in Microbiome	Follow Up
Turner 2020 [100]	UC	16	18-Feb	PUCAI ≥ 65	Vancomycin, Doxycycline, Amoxicillin, Metronidazole	Lower PUCAI levels at antibiotic group	16S RNA in stool	Diversity was reduced. Some patients had higher Escherichia levels after treatment.	12 months
								Recovery after 2 months	
Sporckett 2019 [99]	CD	67	18-May	10 ≤ PCDAI ≤ 40	Metronidazole Versus Metronidazole+ Azithromycin	NS	16S rRNA in stool	Both groups had decreased diversity. Pre-antibiotic microbiome was able to predict response to Metronidazole	12 weeks
						Fcal reduction in combination group			
Levine 2018 [101]		(73 in clinical response analysis)							
Koido 2014 [102]	UC	105 were treated, 12 stool samples analyzed	≥18	Mild to severe UC, with at least 1 relapse a year	Amoxicillin, Tetracycline and Metronidazole	NS	16S	NS	Treatment 2 weeks, follow up for 3 months
							rDNA		
							Real-time PCR quantification of F.Varium DNA		
							in tissue		
Maccaferri 2010 [96]	CD	4	N/A	CDAI > 200	Rifaximin	Not reported	Fecal samples were implemented in colonic models and then analyzed by FISH, qPCR and H-NMR spectroscopy	Increase in concentration of Bifidobacterium, Atopobium and Faecalibacterium prausnitzii.	12 weeks
								Increases in SCFA, propanol, decanol, nonanone and aromatic organic compounds, and decreases in ethanol, methanol and glutamate.	

UC—ulcerative colitis, CD—Crohn’s disease, PUCAI—Pediatric ulcerative colitis activity index, PCDAI—Pediatric Crohn’s disease activity index, CDAI—Crohn’s disease activity index, NS—not significant, Fcal—fecal calprotectin, FISH—fluorescents in situ hybridization, qPCR—quantitative polymerase chain reaction, H-NMR—Hydrogen nuclear magnetic resonance.

**Table 3 ijms-22-12506-t003:** Summary of the microbiome and clinical effects of the different microbiome modulation strategies discussed in this review.

		Crohn’s Disease		Ulcerative Colitis	
Intervention		Children	Adults	Children	Adults
**EEN**	Microbiome effect	↓ *Bacteroides* ↓ *Clostridium Coccoides* ↓ Diversity Butyrate ↓	↓ *Enterobacteriaceae* ↓ Diversity	N\D	N\D
	Clinical response	√	X	X	X
**Prebiotics**	Microbiome effect	↑ *Bifidobacterium*	N\D	↑ *Bifidobacterium*	Bacteroidetes ↑ Butyrate ↑
	Clinical response	√ Protecting factor	√ Protecting factor	N\D	√
**Probiotics**	Microbiome effect	N\D	N\D	N\D	N\D
	Clinical response	X	X	√	√
**Antibiotics**	Microbiome effect	↓ Diversity	*Bifidobacterium* (?) ↑ SCFA (?) ↑	*Escherichia* ↑ ↓ Diversity	No change
	Clinical response	X	X	√ (√√ VEOIBD)	X
**FMT**	Microbiome effect	N\D	Variable study results	N\D	Variable study results
	Clinical response	N\D	X	N\D	X
**Postbiotics**	Clinical response	N\D	√ (?)	N\D	√ (?)

EEN—Exclusive enteral nutrition, N\D—No data, VEOIBD—Very early onset IBD, FMT—Fecal microbial transplantation. X—data suggests no response, √—data suggests response, √√—limited data supports possible increased response in the VEOIBD patient population, ↓—Decreased abundance, ↑—Increased abundance, ?—Questionable effect.

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
