# Peer review of "Therapeutic Advances in Gut Microbiome Modulation in Patients with Inflammatory Bowel Disease from Pediatrics to Adulthood"

_ijms, 2021, doi:10.3390/ijms222212506_

Round 1
Reviewer 1 Report
The manuscript is a narrative review of the gut microbiome of inflammatory bowel disease transitioning from childhood to adulthood. The manuscript offers a good overview of several aspects and is centralized on nutrition. It also indicates the phyla frequently seen. However, as we may know, IBD is a very complex disease with three major types (IBD-Crohn disease, IBD-Ulcerative Colitis, and IBD-U or unclassified). The diagnosis can evolve from IBD-U to Crohn's disease or ulcerative colitis, and the morphology is also variable according to the therapy compliance and nutrition aspects. Therefore, I would suggest the authors emphasize more the hut microbiome change according to the several regimens. Also, the article would greatly benefit if a taxonomic illustration with dendrograms is provided. Dendrograms of phyla would be very useful in the differentiation between nutritional therapies, probiotics, fecal microbiota transplant, and beneficial metabolic derivatives.
Reviewer 2 Report
Remarks to the Author:

Eindor-Abarbanel et al. discuss many microbiome studies of differences and similarities in microbial modulation of pediatric versus adult patients with IBD. With the appreciation of this review, the pediatric and adult patient groups are highlighted. However, I encourage the authors to properly acknowledge many sentences with references; I highly appreciated it to mention original articles rather than reviews. It is quite interesting for section 2.3, however, sections 2.1 and 2.2 are not thoroughly discussed, are incoherent, and include no new information. My comments are attached below.
As stated above, there is no unique perspective in section 2.1. These are available in any standard IBD and microbiome review. The reviewer advises that Section 2.1 can be incorporated into the introduction and that section 2.2 can be combined with 2.3.
Section 2.3 is quite engaging and well-written; this section presents readers with a fresh viewpoint. Please strengthen section 2.3; it would be extremely beneficial to stress the treatment response disparities between pediatric and adult patient groups.
Minor comments:
- Kindly adjust the font size for Line 60, 66, 67, 74, 75, 78, 79, 84, 86, 87, 102, 103, 108, 109, 195, 196, 197, 199, 200, 201, and so on.
- Line 72”. Moreover, under germ-free conditions (no microorganisms) almost all animal models of IBD do not develop disease or disease is significantly attenuated, suggesting that intestinal microbes are essential for the development of intestinal inflammation in IBD”.
Cite the original articles; these claims are not valid; please explain in detail how such claims can be drawn here or later.
- Please add the original citations.
no citations for lines 105-106;
no citations for lines 107-109;
no citations for lines 110-111.
- Line 143-145 “No Metabolite groups of interest include SCFA, bile acids, and tryptophan metabolites, as these metabolites play essential roles in normal immune development, homeostasis, and IBD”.
There was no comprehensive review of the differences and roles of bile acids and tryptophan in pediatric vs adult patients with IBD. While mentioning this is intriguing, it is useful to discuss metabolites in-depth, such as bile acids and tryptophan metabolites, and their differences and roles in pediatric and adult patients with IBD.
Round 2
Reviewer 1 Report
The authors properly addressed the comments and suggestions of the reviewers.
Reviewer 2 Report
There are no additional comments. Thanks.